# The Collapse of Titanium C-Column due to Thermal Compression

**DOI:** 10.3390/ma13184193

**Published:** 2020-09-21

**Authors:** Leszek Czechowski, Adrian Gliszczyński, Nina Wiącek

**Affiliations:** Department of Strength of Materials, Lodz University of Technology, 90-924 Lodz, Poland; adrian.gliszczynski@p.lodz.pl (A.G.); nina.wiacek@dokt.p.lodz.pl (N.W.)

**Keywords:** thermal buckling, post-buckling state, finite element method, structures damage, titanium properties

## Abstract

The analysis of structures under higher temperature is important for predicting the ultimate strength of a structure. Therefore, many experimental tests on samples should be undertaken to observe their behaviour and to determine ultimate load. The present work includes the study on a thin-walled C-column made of titanium compressed in an elevated temperature. The phenomenon of buckling and the post-buckling state of columns were investigated during heating or compressing in higher temperature. The tests of compression were conducted for several temperature increments by assuming the same preload to determine the load-carrying capacity. The deformations of columns until total damage were measured by using the non-contact Digital Image Correlation Aramis^®^ System (DICAS). The numerical calculations based on the finite element method (FEM) were performed to validate the empirical results. The full characteristics of one-directional tension tests were taken into account in order for them to be constant or dependent on the temperature change. Numerical computations were conducted by employing Green–Lagrange equations for large deflections and strains. Based on our own experiment, the thermal property of titanium as a linear expansion coefficient was stable up to 300 °C in contrast to its mechanical properties. The paper shows the influence of varying material properties as a function of temperature on the behaviour and load-carrying capacity of columns. These aspects cause thin-walled columns made of titanium to endure, in elevated temperatures, significantly smaller maximum loads. Moreover, the critical buckling loads for several types of stiff supports were compared to the maximum loads of columns. The results obtained indicate that the temperature rise in columns by 175 K with regard to ambient temperature brings about the decrease of the maximum load by a half.

## 1. Introduction

Titanium alloys are used in special structures and members given their high strength, high-temperature resistance and good corrosion resistance that are outlined in references [1,2]. Moreover, titanium and titanium alloys are among metallic materials applied in the medical and dental branches due to their biocompatibility [3]. They find application as biomaterials in replacing hard tissues carrying heavy loads (bones, teeth, e.g.). Based on thermal compression tests at different temperature it was revealed that titanium alloys due to activation energy can cause thermal deformation [4]. Studies on the behaviour of metal beams in elevated temperature were conducted in references [5,6,7]. The tests of tension on steel samples in high temperature can be found in references [8,9]. Nguyena et al. [10] studied the behaviour of the carbon fibre-reinforced polymer (CFRP) structures subjected to a temperature environment and different mechanical loads. Researchers in papers [11,12] analysed thermal buckling of circular laminated composite or aluminium plates by using the digital image correlation (DIC) technique. In reference [13] based on an experiment performed in elevated temperature it was stated that the behaviour of structures under mechanical loads is strongly dependent on temperature. The authors of paper [14] analysed experimentally and numerically the compression of thin-walled steel C-column in a thermal field. Zhou et al. in reference [15] studied the mechanical properties of CFRP composites at an elevated temperature. Khaneghahi et al. in reference [16] investigated the behaviour of glass fibre-reinforced polymer (GFRP) profiles subjected to compression at higher temperature. Researchers in paper [17] examined the mechanical properties of steel A588 such as Young’s modulus, fracture toughness, yield stress, ultimate stress and surface hardness in temperature up to 815 °C. Zhang et al. in reference [18] studied the mechanical properties of IN718 alloy fabricated by laser metal deposition both in ambient and elevated temperatures. Tests of one-directional tension on metallic samples were conducted in high temperature in references [19,20,21,22], among others. There are many papers based on theoretical approach of thermal analysis on structures whose accomplishments are included in [23,24,25,26,27,28,29]. Theoretical or/and experimental studies involving the pre- and post-buckling state of thin-walled structures were widely discussed in [30,31,32,33,34,35,36,37,38,39]. These works concern mainly the structures built of plane plates where the phenomenon of stability was under investigation. It should be stated that the issue of the present paper is strictly relative to these works by developing a thermal study on thin-walled columns.

In general, referring to the aforementioned literature, one can find many papers where the mechanical material properties in higher temperature for simple samples were obtained. This was owing to the fact that the studies of the stability on thin-walled structures in higher temperature were considered in a few papers. Thus, the verification of behaviour of thin-walled structures in temperature field is still desirable. Therefore, the present paper refers to the analysis of the post-buckling state of thin-walled titanium columns under compression in elevated temperature. Moreover, as was noticed based on accessible literature, titanium structures just in similar conditions have not been investigated yet. In this work, the edges of columns were put into grooves of special plates to realize supports corresponding or comparable to articulated supports. The tests of compression on columns lasted until peaks of loads were reached to determine the curves of work and the load-carrying capacity of studied columns. In addition, the Digital Image Correlation Aramis^®^ system (DICAS) [40] suitable for registrations of deformation maps was used. In numerical computations, full characteristics of titanium attained due to one-directional tension tests at ambient temperature and at higher temperature were taken into account. Results revealed that both the behavior and load-carrying capacities of titanium columns are strongly linked to the thermal environment. This aspect seeming to be obvious should be important because titanium is regarded as the high-temperature resistant material but mechanical properties of titanium decrease considerably with an increase of temperature. This fundamental feature can play an essential role as far as structures made of titanium are considered to be loaded in elevated temperatures. Furthermore, as was observed, soft titanium structures in a vicinity of supports in elevated temperatures can deform easily. This phenomenon can cause locally initial imperfections to appear in structures, which can have an influence finally on the significant drop of the load-carrying capacity of columns. 

## 2. Problem Description

### 2.1. The Study Object

A thin-walled C-profile column with dimensions of L = 250 mm, b = 80 mm, a = 40 mm and thickness t = 1 mm under compression in elevated temperature was investigated (Figure 1). The tests of the compression were carried out to determine the whole curves of load vs. shortening. The columns were made of titanium (type: grade 2, content of elements: T > 98.9%, Fe < 0.3%, O < 0.25%, C < 0.08%).

The tests of compression were executed in thermal chamber (Instron, model 3119-605) by means of an Instron machine-model 5982 (Figure 2). This machine enables compression tests in range from 0.02 N to 100 kN to be conducted. The employed chamber allows to reach maximum temperature coming to up 350 °C. During tests, the deformations of samples were registered by using non-contact DICAS [40] equipped with two cameras set vertically. The columns assigned to tests were painted by applying the high-temperature resistant paint.

The experimental study was composed of three stages. The first stage involved a initial load of sample (circa 5–10% of maximum load). The second stage was based on the temperature increase by appropriate temperature increment (dT) where grips of the machine remained unmoveable (this then caused the reaction forces to increase). The heating process of columns took usually about 2 h. The last step related to compression in elevated temperature to determine the post-buckling equilibrium paths and the peaks of loads (the load-carrying capacity). 

### 2.2. Material Data

The conducted one-directional tensile tests enabled to determine material characteristics in whole range both at ambient temperature and at temperature of 100 °C and of 200 °C (Figure 3a). 

For numerical simulation, data of these curves were transformed to curves in a function of true stresses vs. logarithmic strains. By linear interpolation, more characteristics of tension dependent upon temperature were obtained. Based on this modification, all curves shown in Figure 3b were implemented to computation software. The mean Young’s modulus and mean Poisson’s ratio were assumed to be independent on temperature and equal 110 GPa and 0.28, respectively. Based on measures of strains by using DICAS in a range from 25 °C to 200 °C in the thermal chamber, the thermal expansion coefficient of titanium was determined ($\alpha_{Ti}=8.8^{-6}{}_{}\;1/K$) which turned out to be stable in analysed scope. Subsequently, this value was taken in simulations.

### 2.3. Finite Element (FE) Model

Numerical simulations were performed based on the finite element method with the use of NASTRAN/MARC FEA 2010R^®^ version software [41]. A discrete model of columns was generated due to extruding the first-order shell finite elements to create the first-order solid finite elements (Figure 4). The number of elements throughout the thickness of column was equal 4. Then, the number of solid elements in the sample was 100,000. However, the total number of elements used in analysis was above 260,000. Numerical calculations were carried out by activating Green–Lagrange equations for large strains and deflections as well as the second Piola–Kirchhoff stress [41]. The number of sub-steps for each simulation was set to be between 100 and 50,000. To ensure adequate convergence, the number iterations of each sub-step included from 10 up to 5000. The simulation was divided to three stages (three load-steps) to reflect conditions as in experiment. The load-step-1 (with boundary conditions BC-1) involved the small shortening ($U_{y}$) of the columns to activate the contact elements between discrete models (way of the column support-Figure 5). The friction coefficient was set to 0.05. The contact detection parameter based on bias on tolerance was assumed. The separation criterion on forces was taken into account. The load-step-2 was based on increasing temperature by some temperature increment (with boundary conditions BC-2). Both plates keeping column on their external surfaces were unmoveable (it means that $U_{x}=U_{y}=U_{z}=0$). The load-step-3 (BC-3) was continuing load-step-1 at the presence of BC-2 (in thermal field) to completely compress the column (until the load-carrying capacity was determined). The boundary conditions of supports are given in Figure 4. In simulations, only perfect columns were taken into consideration. 

## 3. Results

### 3.1. Buckling Forces

This subsection includes the results of critical loads of the column (Figure 6). Therefore, separate analyses of linear eigenvalue buckling were performed to realize similar conditions of supports as illustrated in Figure 5. It was necessary to refer the estimations of buckling loads to the load peaks. Because it was hard to define possible contact relations during compression between column and plate which can occur, four different types of stiff support were considered. The first type S_1 involved the articulated support on mid-lines of outer surfaces. The second type S_2 included the fully clamped outer surfaces of the column. The conditions of supports S_3 and S_4 referred to a simply supported column such as S_1 but either on the outer (S_3) or inner edges of the columns (S_4). The two last conditions of support were considered to verify the influence on the buckling loads because in the case of certain tolerance of dimensions in the studied column, these methods of support might be possible. Critical loads of columns were only numerically defined. The estimated buckling loads are described as: $F_{cr}^{S\_1}$- S_1, $F_{cr}^{S\_2}$- S_2, $F_{cr}^{S\_3}$- S_3, $F_{cr}^{S\_4}$- S_4. The buckling temperature increments and their relating reaction forces are denoted as: $\Delta T_{cr}^{S\_1}$- S_1, $\Delta T_{cr}^{S\_2}$- S_2, $\Delta T_{cr}^{S\_3}$- S_3, $\Delta T_{cr}^{S\_4}$- S_4 and $F_{Rcr}^{S\_1}$- S_1, $F_{Rcr}^{S\_2}$- S_2, $F_{Rcr}^{S\_3}$- S_3, $F_{Rcr}^{S\_4}$- S_4, respectively. All these values are shown in Table 1 (Temp represents critical values for temperature increments and Comp denotes the critical loads due to compression). 

Critical reaction forces were determined by multiplying critical temperature increment, Young’s modulus, the cross-section area of the column and the thermal expansion coefficient (formula based on Hook’s law). Of course, the assumed types of support (S_1, S_2, S_3 and S_4) are pretty different from those realized in experiments because the articulated support of columns would be hard to elaborate. In the case of the determination of the critical temperature increments, the uniform temperature increase in the entire structure of the column was taken into account. The critical values were computed for the first five buckling loads. Based on FEM results, differences in critical forces in the case of thermal buckling are meaningful (first critical force ranged from $F_{Rcr}^{S\_1}=7431.8^{}\;N$ for S_1 to $F_{Rcr}^{S\_4}=9793.0^{}\;N$ for S_4). Hence, there is an increase by 31.8%. By referring to critical forces due to compression, the discrepancies for different variants of support are significantly minor, but the lowest critical force was observed for condition S_1 and the highest critical force was obtained for condition S_2 (clamped surface). This trend is kept also for critical loads for higher modes. By analysing the modes of detailed variants, two half-waves along the axis of column were noticed (Temp, S_1, S_2, S_3 and S_4) however for S_3 and S_4 (Comp) two half-waves were not full (See Table 2). From the physical point of view, the observed number of half waves should not be related to the values of buckling loads based on different supports because different lengths of half-waves are possible, but critical loads can differ from each other (compare mode of S_1 with mode of S_2 for Temp). Moreover, the noticed modes of columns in linear buckling analysis for a given support can often be regarded as modes of the initial deformations during compression but one should underline that during full compression the modes of deformations in columns with respect to applied load can vary. However, in the case of deformation modes for higher critical loads, numbers of half-waves range from 3 to 4. 

### 3.2. Compression in Ambient Temperature

Figure 7 presents the charts of compression force F vs. shortening Δ for experiment and numerical method. The maximum load obtained in the FEM result is close to maximum load recorded in the experiment (there are three experimental curves).

The noticed difference in maximum loads amounts to 1–2% ($F_{max}^{FEM}=20.1^{}\;kN$ and $F_{max}^{EXP}=20.4^{}\;kN$-mean value). The maximum load is about two times greater than buckling load as it is given in Section 3.1. The visible discrepancies between curves correspond to full shortening of the column (maximum load in FEM appears considerably earlier). The shapes of deformation for selected points of load during compression were also included in Figure 7. A visible buckling (three half-waves) was noticed between the 8 kN and 10 kN points (b). Three half-waves lasted until maximum load was reached at points (c) and (d). Afterwards, in the lower part of the column, large deformations followed which caused damage to the structure point (e). 

### 3.3. Thermal Compression

This subsection presents the results of the experiment and numerical approach. Figure 8 shows the reaction forces *F_R_* vs. the shortening of column d_T_ in elevated temperature. The tests of the columns behaviour were carried out for seven different temperature increments dT (from 25 K to 175 K every 25 K) relative to ambient temperature. Compression of columns for each considered temperature increment for two samples was performed (except for 175 K). The detailed tests were denoted: EXP_1_1 and EXP_1_2 for dT = 25 K, EXP_2_1 and EXP_2_2 for dT = 50 K, EXP_3_1 and EXP_3_2 for dT = 75 K, EXP_4_1 and EXP_4_2 for dT = 100 K, EXP_5_1 and EXP_5_2 for dT = 125 K, EXP_6_1 and EXP_6_2 for dT = 150 K, EXP_7_1 for dT = 175 K. By taking into consideration the maximum loads, some decrease of load-carrying capacity with an increase of temperature can be observed. The maximum mean load for DT = 25 K amounted to 19.1 kN (decrease by circa 6% with respect to load-carrying capacity in ambient temperature). The maximum mean load for other temperature increments dT = 50 K, dT = 75 K, dT = 100 K, dT = 125 K, dT = 150 K and dT = 175 K amounted to 17.3 kN, 16.3 kN, 15.1 kN, 13.6 kN, 10.9 kN and 10.2 kN, respectively. It can be clearly noted that the mechanical properties of titanium are quite strongly temperature-dependent even for small temperature increase. Figure 9a illustrates the curves of reaction forces in a function of temperature increase for FEM. For small temperature, many curves of analysed variants overlap with each other and they cannot be visible. The last points of curves from Figure 9a are the beginnings of curves depicted in Figure 9b. Moreover, the curves almost run very close regardless of assumption of stable or temperature-dependent mechanical properties. Of course, the lowest load-carrying capacity was noticed for the highest temperature increment dT = 175 K (then, the maximum load amounted to 12.5 kN) but already during heating (BC-2). Indeed, this value is very close to maximum load achieved in the experiment but in the case of the experiment the maximum load occurred during just compression (BC-3). In general, it should be emphasised that in contrary to the experiment, higher reaction forces were collected in the FEM. 

Based on curves for stable mechanical properties of columns (temperature-independent), the load-carrying capacities remain almost on this same level regardless of the temperature increase (this value amounts to 20.6–21.5 kN). In addition, it can be noticed that all these values are higher than the maximum load obtained during compression in ambient temperature. By taking into consideration the temperature-dependent mechanical properties, the attained characteristics approach the experimental curves (a comparison of some curves attained by two methods is displayed in Figure 10). However, the numerical curves of the compression are definitely shorter (the maximum loads are reached faster) than in the case of the experimental curves. Apparently, such a trend can be caused due to an assumption of a perfect column in the numerical model because the field of occurring plastic deformations is focused on some parts of the column (See Figure 7d–e). In a comparison with the experiment, estimated maximum load based on simulation is greater by 2.6% for dT = 25 K and greater by 38.2% for dT = 150 K (for this variant difference is the biggest). The maximum loads for all variants are given in Table 3.

Table 4 shows the modes of total displacements based on DICAS and FEM results. The maps are presented for all considered temperature increments. Point 1 denotes the maps of deformations of columns after heating. Point 2 represents the modes during reaching the maximum load (beyond the last variant because the maximum load was obtained only during heating). The last point, point 3, presents maps of columns with large deformations at the end of the tests. In general, the obtained deformations are similar. Of course, contrary to the experiment, in the case of FEM half-waves along the axis of columns are meaningfully visible (especially in the case of point 1). More noticeable similarities are seen for point 2 (by comparing the maps in maximum load). In the case of FEM, three half-waves are observed (in the experiment either two half-waves or three half-waves are observed). The maps of damage (point 3) are comparable; however, in the experiment for some cases mechanisms of failure occurred in the middle of columns (dT = 25 K*,*
dT = 50 K, dT = 125 K) but in the case of FEM places of damage appeared always in the vicinity of one of the supports.

## 4. Summary

This paper includes the result of the study of compressed thin-walled columns at elevated temperature. The experimental and numerical investigations were performed to examine the behaviour of titanium columns supported in grooves of plates. Based on the results, the main conclusions that can be drawn are:Both mechanical properties and load-carrying capacity of titanium columns are very sensitive to temperature rise although the titanium is resistant moderately to high temperature. The increase of temperature increment by 175 K referring to ambient temperature causes the load peak during compression to decrease almost by a half. Moreover, as was observed, titanium columns under load in elevated temperatures became slightly mild which might bring additional local deformation diminishing the general load-carrying capacity of columns.The assumption of stable mechanical properties of titanium (temperature-independent) in numerical simulations does not allow proper load-carrying capacity to be achieved which can be compared to experiment. In the case of consideration of stable mechanical properties of titanium, obtained load peaks for higher temperatures are almost the same.Regarding buckling loads, considered boundary conditions S_3 and S_4 (Temp) gave the highest critical forces (almost 10 kN). The visible clues of appearing half-waves in columns during experiments were noticed to be also in a range of loads of 8–10 kN. Moreover, estimated critical loads were circa two times smaller than the load-carrying capacity of the column in ambient temperature.The maximum loads obtained numerically were pretty close to experimental ones although the discrepancies amount up to 40%, at most (for temperature increment 150 K). It can be explained that the analysed phenomenon of full behaviour of thin-walled columns in a thermal field seems to be very complex. Firstly, studied columns could be imperfect resulting from a tolerance of thicknesses, other dimensions and the whole shape. Secondly, mechanical properties of titanium are very dependent on temperature changes. Thirdly, the steel plates (upper and lower) with made channels (grooves) holding the columns possess a proper thermal expansions coefficient, which is slightly different from the thermal expansion coefficient of titanium. Therefore, during compression the conditions of supports could continuously change. Furthermore, in numerical simulations, some simplifications (linear interpolations of one-directional tensile curves for adequate temperature) were applied to solve the problem in an approximate way.The curve of force vs. shortening of the column for ambient temperature for numerical model is comparable in reference to experimental curves but in the case of numerical model, the obtained range of shortening in column was shorter. This could result from the fact that numerical model was assumed as perfect what brings locally to concentrations of maximum plastic stresses after the load peak. Therefore, the effect could cause a drop of load to follow violently.By comparing DICAS and FEM maps, the modes of deformation in all phases seem usually to be alike. Both greater discrepancies and completely different modes can be observed as well. It can be clarified that each column under compression in elevated temperature can deform in different way.The appearance of mechanisms (large deformations) of the columns during the last stage of complete damage occurred in the vicinity of that of support (especially in the case of numerical results). In the case of some samples, failure of the columns occurred in the middle (dT = 25 K, dT = 50 K, dT = 125 K).

## Figures and Tables

**Figure 1 materials-13-04193-f001:**
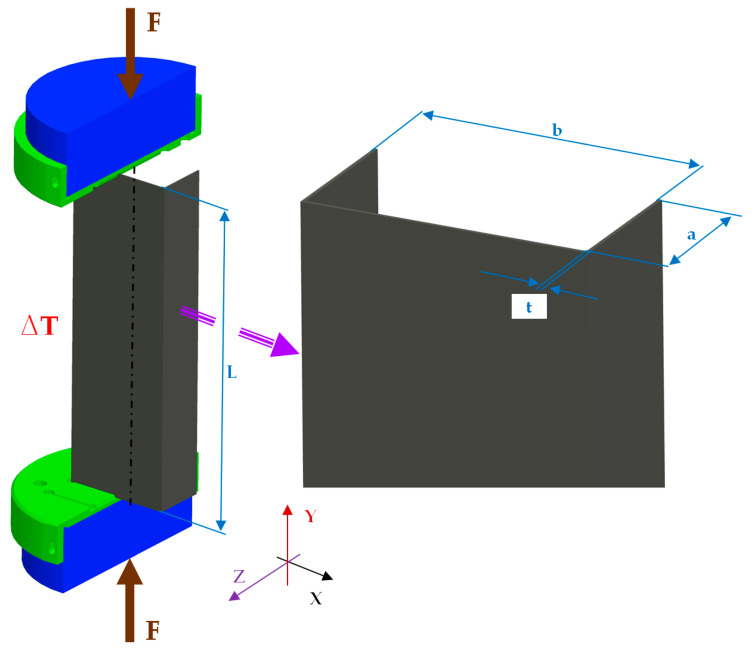
The object of the study.

**Figure 2 materials-13-04193-f002:**
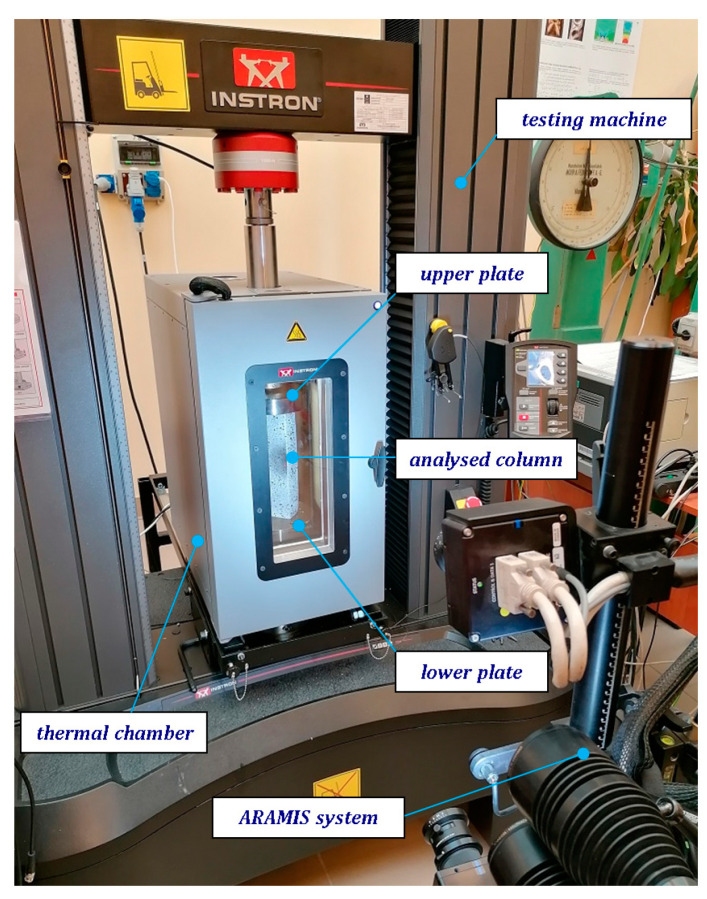
The testing stand equipped with the Aramis^®^ (digital image correlation) system.

**Figure 3 materials-13-04193-f003:**
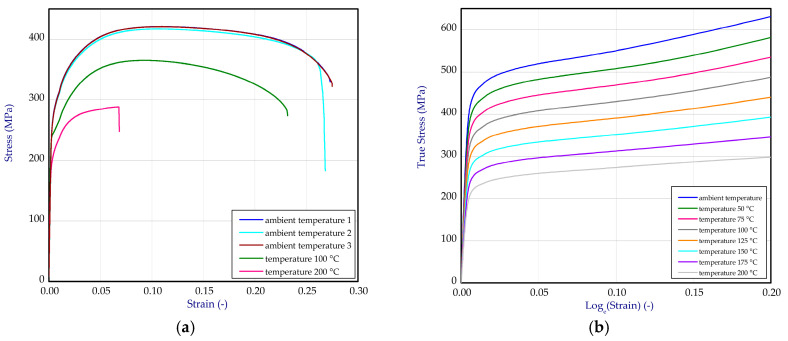
The tensile characteristics of titanium (**a**) and curves after transformation into true strain-stress relation (**b**).

**Figure 4 materials-13-04193-f004:**
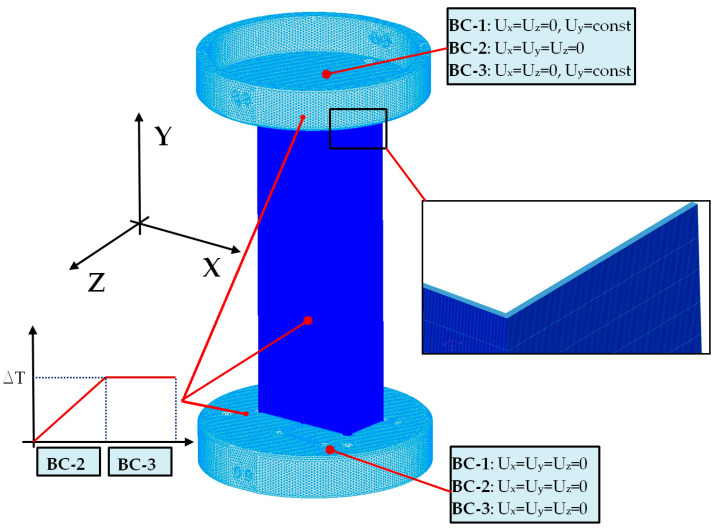
The finite element (FE) model with a description of boundary conditions.

**Figure 5 materials-13-04193-f005:**
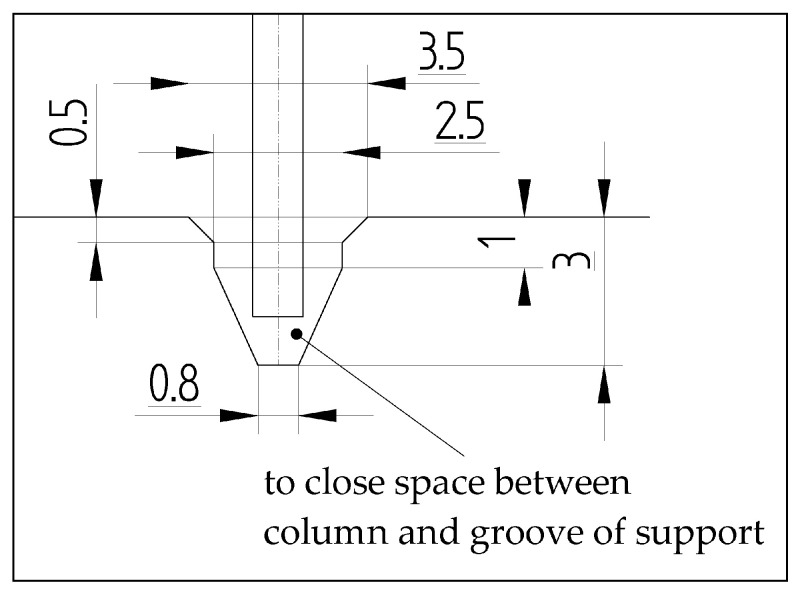
The way of support of column edges in the groove of the plate.

**Figure 6 materials-13-04193-f006:**
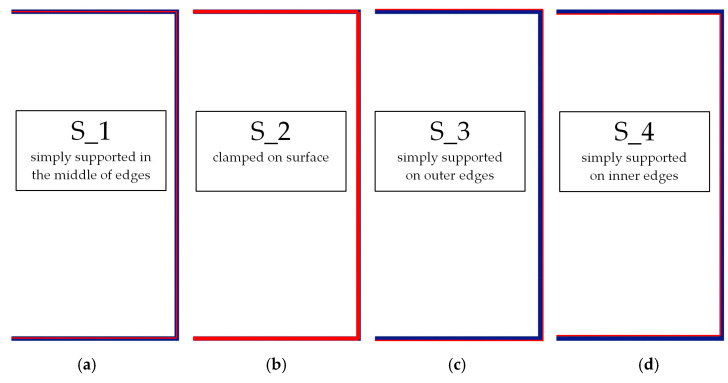
The type of upper and lower supports of column for (**a**) S_1 (**b**) S_2 (**c**) S_3 and (**d**) S_4.

**Figure 7 materials-13-04193-f007:**
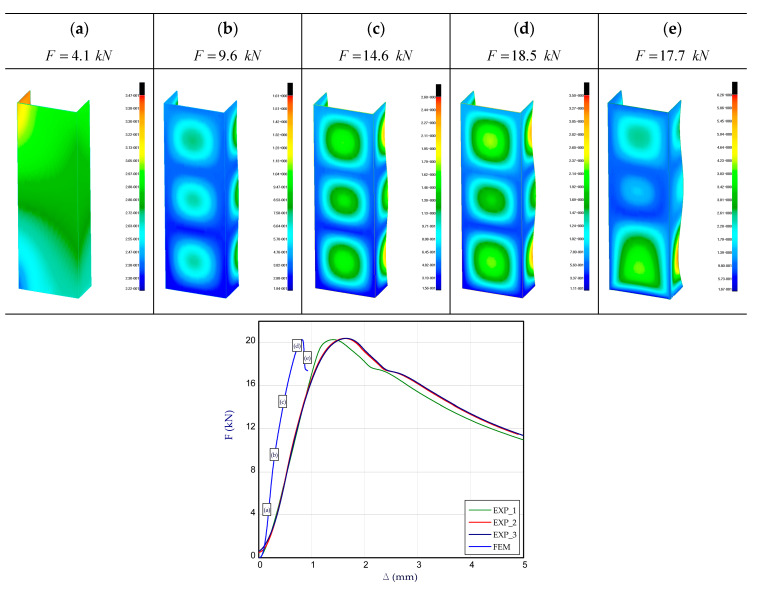
The compression curve (force vs. shortening) of column with deformation maps of column: (**a**–**e**).

**Figure 8 materials-13-04193-f008:**
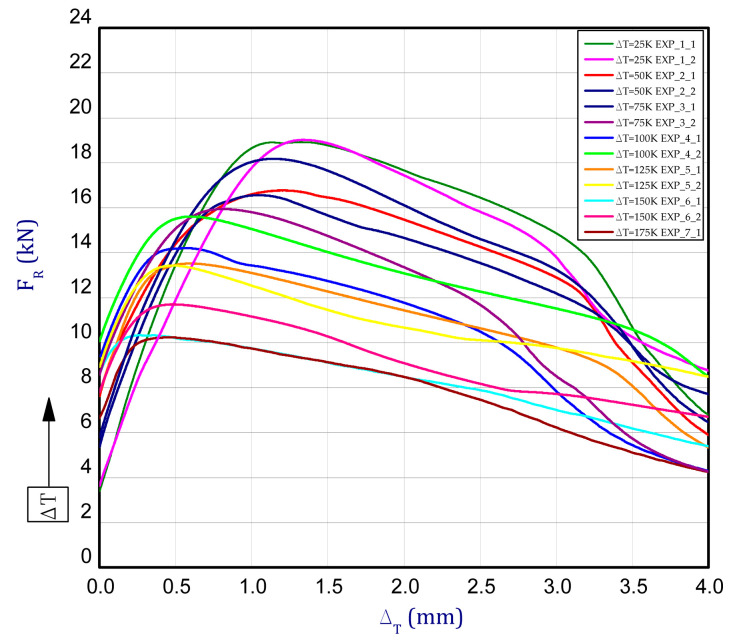
The reaction force vs. shortening of column for several temperature increases.

**Figure 9 materials-13-04193-f009:**
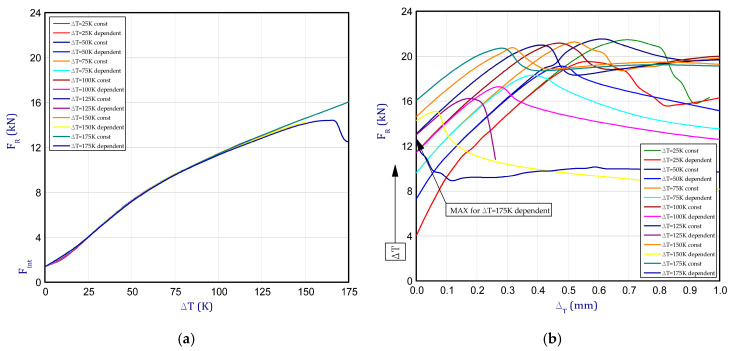
The force reaction vs. (**a**) the temperature increment (BC-2) and (**b**) vs. shortening of column (BC-3) based on finite element method (FEM).

**Figure 10 materials-13-04193-f010:**
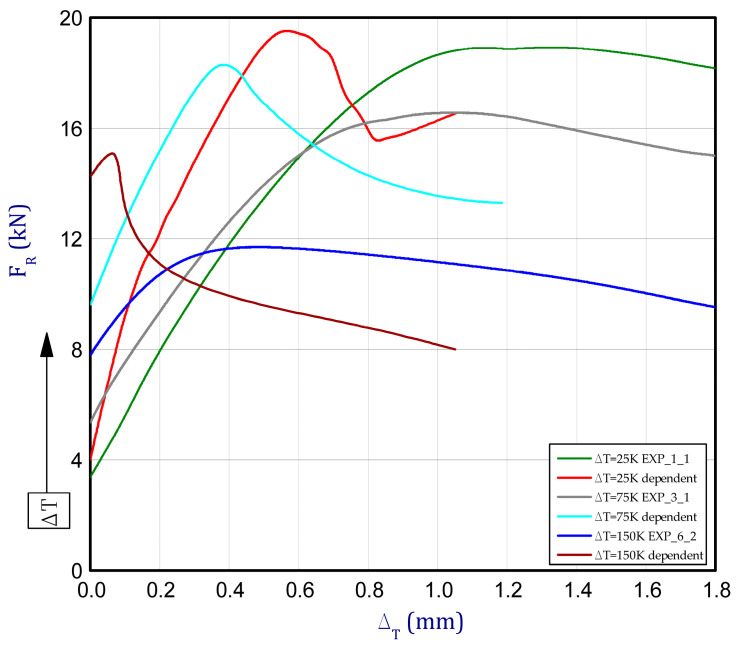
Comparison of obtained curved by FEM and EXP.

**Table 1 materials-13-04193-t001:** The critical loads with different boundary conditions.

Mode	Temp S_1 $\Delta T_{cr}^{S\_1}$ [K] $\left(F_{Rcr}^{S\_1}\right)$ [N]	Temp S_2 $\Delta T_{cr}^{S\_1}$ [K] $\left(F_{Rcr}^{S\_2}\right)$ [N]	Temp S_3 $\Delta T_{cr}^{S\_3}$ [K] $\left(F_{Rcr}^{S\_3}\right)$ [N]	Temp S_4 $\Delta T_{cr}^{S\_4}$ [K] $\left(F_{Rcr}^{S\_4}\right)$ [N]	Comp S_1 $F_{cr}^{S\_1}$ [N]	Comp S_2 $F_{cr}^{S\_2}$ [N]	Comp S_3 $F_{cr}^{S\_3}$ [N]	Comp S_4 $F_{cr}^{S\_4}$ [N]
1	49.1 (7431.8)	52.0 (7870.7)	64.0 (9687.0)	64.7 (9793.0)	7546.8	8110.1	7807.4	7759.0
2	50.1 (7583.1)	55.5 (8400.5)	65.4 (9899.0)	68.6 (10383.3)	7774.7	8650.1	8541.3	8463.5
3	60.4 (9142.1)	68.2 (10322.8)	80.9 (12245.0)	82.5 (12487.2)	9194.2	10564.93	9969.8	9903.8
4	64.3 (9732.5)	69.2 (10474.1)	85.8 (12986.7)	85.7 (12971.6)	9850.6	10735.3	10421.1	10302.2
5	69.7 (10549.8)	72.7 (11003.9)	92.1 (13940.3)	91.6 (13864.6)	10672.0	11411.1	11643.8	11426.1

**Table 2 materials-13-04193-t002:** The buckling modes of the columns.

Number of Mode	Type of Boundary Conditions							
	Temp S_1	Temp S_2	Temp S_3	Temp S_4	Comp S_1	Comp S_2	Comp S_3	Comp S_4
1	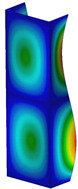	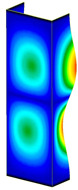	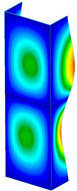	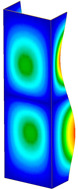	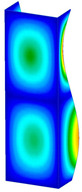	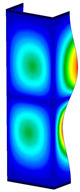	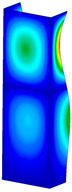	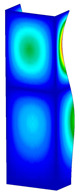
2	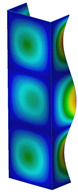	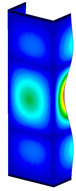	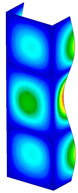	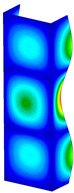	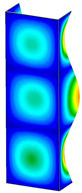	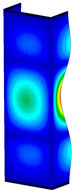	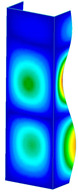	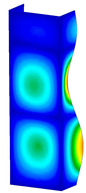
3	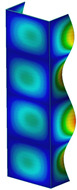	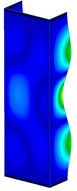	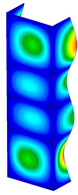	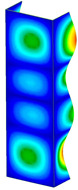	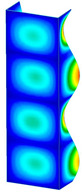	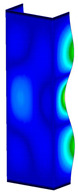	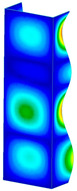	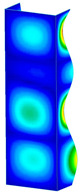
4	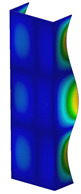	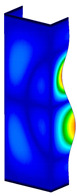	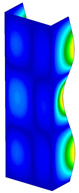	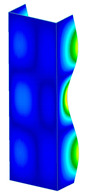	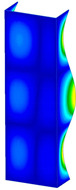	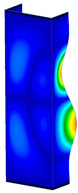	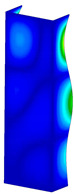	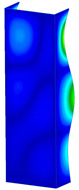
5	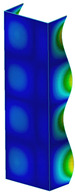	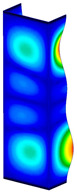	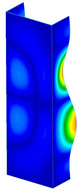	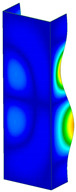	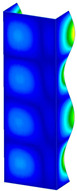	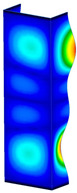	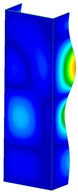	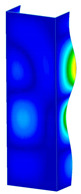

**Table 3 materials-13-04193-t003:** The load-carrying capacity based on FEM and the experiment (EXP).

Temperature Increment [K]	FEM Independent [kN]	FEM Dependent [kN]	EXP			Increase of Maximum Load (FEM-Dependent) with Regard to Experiment [%]
			Attempt 1 [kN]	Attempt 2 [kN]	Mean Value [kN]	
25	21.5	19.5	18.9	19.1	19.0	2.6
50	21.6	19.2	16.7	18.1	17.4	10.3
75	21.4	18.4	16.5	16.1	16.3	12.9
100	21.2	17.4	14.2	15.6	14.9	16.8
125	21.0	16.3	13.5	13.3	13.4	21.6
150	20.9	15.2	10.3	11.7	11.0	38.2
175	20.8	12.5	10.2	-	10.2	22.6

**Table 4 materials-13-04193-t004:** The comparison of recorded modes in total displacements for both methods.

	DT = 25 K		DT = 50 K		DT = 75 K		DT = 100 K		DT = 125 K		DT = 150 K		DT = 175 K	
	FEM	DICAS	FEM	DICAS	FEM	DICAS	FEM	DICAS	FEM	DICAS	FEM	DICAS	FEM	DICAS
1	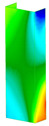	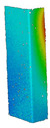	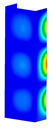	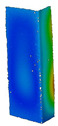	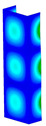	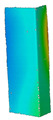	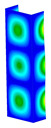	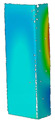	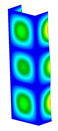	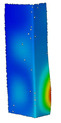	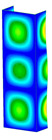	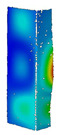	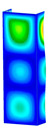	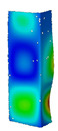
2	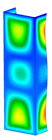	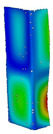	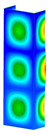	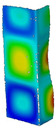	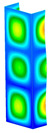	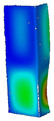	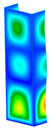	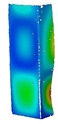	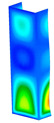	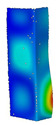	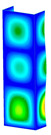	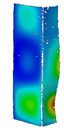	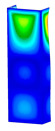	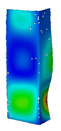
3	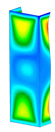	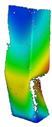	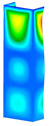	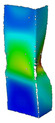	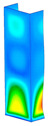	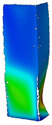	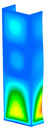	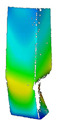	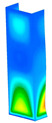	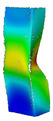	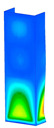	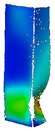	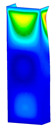	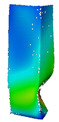

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
