# Peer review of "The Collapse of Titanium C-Column due to Thermal Compression"

_materials, 2020, doi:10.3390/ma13184193_

Round 1
Reviewer 1 Report
This paper investigates the compressed thin-walled columns till full damage in elevated temperatures. The experimental procedures and numerical calculations are both well performed. Before it can be published, some minor revisions are needed:
1. Please write the Section 2.1 into the Section 2.2.
2. If the authors want to point out the importance of each table or figure, the summary should be written as one complete paragraph instead. Please rewrite the summary.
Author Response
Dear Reviewer,
we’d like to thank for review and your valuable remarks. We have done our best to improve our paper and to fulfil the requirements of publishable standard.
Best Regards,
Leszek Czechowski
Adrian Gliszczyński
Nina WiÄ…cek

Reviewer 2 Report
This paper studies the mechanical behaviors of titanium thin-walled C-column under compression test in different temperatures. Both numerical simulation and experiments are employed to understand this behavior. The result shows the temperature increment has great influence on loading-carrying capacity of titanium. However, this paper has low quality with seldom novelty. This decision is based on the following critical issues.
- The present paper contains countless ill constructed sentences that make it quite demanding for readers to understand. The writing is not concise and informative, requiring substantial improvement. Especially, the misusing of ‘rely upon’,‘depend on’ and stuff make it very challenging for readers to understand information behind this paper.
- There is no figure (b) in the title of Figure 3. Meanwhile the number of curves in Fig.3(b) is more than the Fig.(a), which means that some data are not presented in the Fig.3(a). The author should complement related data to ensure the results more reasonable.
- In 116 and 117 lines, there is no boundary conditions mentioned in Fig.3
- The author makes many basic errors like numbering wrongly the table numbers, which makes it difficult for the present reviewer not to doubt the authors’ serious attitude towards the present work. The reviewer is not happy to read the unpersuasive expressions in almost of the whole paper.
- About the FE model, the author needs to describe the constitutive equation for finite element simulation in detail since it is a very important part and determines the reliability of the simulation results. The authors should provide the crucial information that may determines the final results. For instance, the simulation results are not well fitted to the experimental works (Fig. 7). The underlying reason could be due to the lack of constitutive equation for the investigated titanium.
- When the authors discussed the simulated results from the different modes, he did not explain the physic meanings and causes of formation of half-waves. So that it is very puzzled about the discussion of results. And it is incapable for the readers to understand further the purposes of carrying out simulation. This looks more like a technical report, rather than a scientific paper!
- In the thermal compression and compression in ambient temperature, the authors put experiments results and FEM results together. However, it is unable to comprehend the links between them relied on current expressions. Since these two parts are the core contents of this paper, the authors should clarify their logic and use more understandable and readable way to express the novelty and scientific values of this work. The present paper failed on these aspects.
Based on the above considerations, the present reviewer does not recommend this paper for publication in Materials.
Author Response

(The authors gave the same response as above.)

Reviewer 3 Report
This work is interesting; however the presentation style is poor, especially related to many contradictions. Please revise it accordingly. Here also some comments which can helps
The novelty of this work is not very clear as there also are numerous publication, even the authors claim this “ there are few papers”…so which are the benefits of state of art using this work ?
Some English requires adjustments (I.e “Titanium thin-walled C-profile column with dimensions of L = 250 mm, b = 80 mm, a = 40mm and thickness t = 1 mm under axial compression both during temperature increase and in elevated stable temperature till the damage », an dother pharses found teh same
Please provide details of materials analysed cause you suggest the content is almost as pure titanium, but what are materials properties for pure titanium ?
« of samples were watched by non-contact DICAS « the deformation were watched or measured ?
Please provide details of « temperature increment (DT). »
« Based on these curves, data were transformed to true stresses - logarithmic strains which were
implemented to software (Fig. 3b). » Howvere in Figure 3a you have 5 curves and Figure 3b you have 8 curves how did you transformed it ?
« At greater temperature increments used in simulations, the load-step-3
didn’t occur with respect to earlier appearing maximum load (during heating up). » do not make sense
« The assumed types of supports (S_1, S_2, S_3 and S_4) are pretty different from
realized in experiment because the articulated support of columns would be hard to elaborate (See
Fig. 5). » and what was teh emaning then ? to have differnt simulation.
Why the forces do not start from 0 and starts from 1, 2.5, 4 KN in Figure 8, 9, 10.? There was a problem of calibration ?
“numerically are pretty close to experimental ones. The discrepancies amount up to 40 %, at most (“please check this statement, as close do not means differences up to 40%
The conclusions should be reformulated as they as very contradictory
Author Response

(The authors gave the same response as above.)

Round 2
Reviewer 2 Report
It seems that the authors of this submission have made sufficient corrections to secure a publication in Materials. I have no other comments for this manuscript.
Reviewer 3 Report
.